# “Know Your Children, Who They Are, Their Weakness, and Their Strongest Point”: A Qualitative Study on Diné Parent Experiences Accessing Autism Services for Their Children

**DOI:** 10.3390/ijerph20085523

**Published:** 2023-04-14

**Authors:** Olivia J. Lindly, Davis E. Henderson, Christine B. Vining, Candi L. Running Bear, Sara S. Nozadi, Shannon Bia

**Affiliations:** 1Department of Health Sciences, Northern Arizona University, Flagstaff, AZ 86011, USA; 2Department of Communication Sciences and Disorders, Northern Arizona University, Flagstaff, AZ 86011, USA; 3A.T. Still University, Mesa, AZ 85206, USA; 4Department of Educational Specialties, Northern Arizona University, Flagstaff, AZ 86011, USA; 5Health Sciences Center, University of New Mexico, Albuquerque, NM 87131, USA; 6National University, San Diego, CA 92123, USA

**Keywords:** autism spectrum disorder, diagnosis, treatment, services, Diné, Navajo, Indigenous, Southwest, United States

## Abstract

Background and Objective: Marked inequities in access to autism services and related health outcomes persist for U.S. children, undermining broader initiatives to advance the population’s health. At the intersection of culture, poverty, and ruralness little remains known about autism in many Indigenous communities. This qualitative study on the lived experiences of Navajo (Diné) parents raising a child with autism sought to identify factors affecting access to services. Methods: A Diné researcher conducted in-depth interviews with 15 Diné parents of children with autism living in or around the Navajo Nation. A directed content analysis approach was used to identify themes, subthemes, and connections between themes. Results: Twelve overarching themes emerged on Diné parents’ experiences accessing autism diagnostic and treatment services, as well as ways access to autism services can be improved. The following themes were related to diagnosis: the diagnostic process was often emotionally fraught; long wait times of up to years for diagnostic services were commonplace; limited clinician training and cultural humility impeded access to diagnostic services; and adequate health insurance, Indian Health Service referrals, care coordination, financial aid for travel, and efficient evaluation facilitated diagnosis. Themes on treatment access were as follows: parent perceptions of the extent to which an autism service helped their child affected access; social support helped parents to access treatment; obtaining referrals and care coordination influenced treatment access; treatment costs affected access; and service availability and geographic proximity impacted treatment access. Themes on ways to improve access to autism services were as follows: greater autism awareness is needed; autism-focused support groups may be helpful; and increased availability and quality of autism services across and around the Navajo Nation is paramount. Conclusions: Diné parents’ access to autism services was dynamically affected by sociocultural factors that must be addressed in future health equity-oriented initiatives.

## 1. Introduction

Autism spectrum disorder (hereinafter referred to as autism), which is characterized by challenges with social communication and restricted and repetitive behaviors [1], has become increasingly prevalent among children [2,3]. According to the Centers for Disease Control and Prevention and Developmental Disabilities Monitoring Network, an estimated 1 in 44 children aged eight years old have autism [2,3]. Today, early autism diagnosis before a child turns three years old is possible and has increased [3,4]. Early diagnosis and treatment with evidence-based services (e.g., early intensive behavioral intervention) may help to optimize health for children with autism and their families [5]. However, late diagnosis and difficulty accessing needed treatment services (hereinafter collectively referred to as autism services) are commonplace [6,7]. For underserved children—especially those belonging to racial and ethnic minority groups, whose families are low income, and/or who live in rural areas—disparities in access to and quality of autism services are pronounced and persistent [8]. Disparities in access to autism services have been established for children who are Latino [9], Black or African American [10], Native Hawaiian or Pacific Islander [11], and American Indian or Alaska Native (AI/AN; hereinafter referred to as Indigenous) [12,13]. For example, culturally and linguistically diverse individuals are not always accurately identified with autism (i.e., under- or over-identification), which may contribute to inequitable access to services such as behavioral therapy that promote child development and functioning [14]. Little research has, however, been conducted to better understand why persistent inequities in autism services access exist for Indigenous children including those who identify as Navajo (hereinafter referred to as Diné, meaning “The People”)—the largest federated tribe in the United States.

The profound racial and ethnic disparities in access to autism services that exist broadly for culturally and linguistically diverse children are generally theorized to stem from the intersections between the sociocultural factors within communities (e.g., traditions, values, beliefs) and the structural factors within systems of care such as health insurance policy and limited workforce diversity [15,16]. Cultural values held by culturally and linguistically diverse parents and guardians (collectively referred to as parents/guardians hereafter) about children with autism (e.g., parent/guardian beliefs that a child’s autism is not a disability) may, for instance, influence their perceptions of the child’s service needs and the child’s response to treatment [17]. Although little research has examined Diné cultural beliefs specifically about autism, Vining and Allison conducted a study to better understand Diné beliefs on development and disability more broadly [18]. This research described how Diné families of children with developmental disabilities may believe that there are cultural and spiritual explanations for their child’s disability [18]. Elders warn expecting parents to guard against certain exposures because they could lead to disabling conditions or symptoms (e.g., cleft lip/palate could be associated with a parent engaging in silversmithing because it requires cutting metals and other elements). These, along with other cultural beliefs, may motivate parent/guardian decisions to seek traditional healing (e.g., a purification ceremony) to restore harmony and ensure the child’s well-being rather than mainstream, commonly recommended autism services such as applied behavior analysis (ABA). Diné families may also believe that having a child born with a disability is a gift from the creator and that they should care for and teach their child so the child can be a part of the family and community and live a meaningful and productive life [18]. These types of sociocultural factors along with structural factors are part of systems of care including market failures (e.g., limited autism services availability in rural areas), restricted pathways of care (e.g., poor family–provider interaction), and negative clinical encounters (e.g., limited clinician training on autism and cultural humility—“a lifelong commitment to self-evaluation and self-critique, to redressing the power imbalances in the patient–physician dynamic, and to developing mutually beneficial and non-paternalistic clinical and advocacy partnerships with communities on behalf of individuals and defined populations” [19]) often result in inequitable access to autism services and cumulative disadvantages for children and their families [15]. The Modified Sociocultural Framework for Autism Health Services Disparities demonstrates these relationships (Figure 1; [15]).

An estimated 28.5 percent of individuals are Indigenous in Northern Arizona [20], and Diné individuals comprise the largest segment of the Indigenous population in this geographic area [21,22]. As the largest federated tribe, the Navajo Nation has an estimated 165,158 Diné individuals living within its boundaries spanning Arizona, New Mexico, and Utah (Figure 2; [23]). The Diné are historically nomadic, with clusters of homes scattered throughout rural regions in small towns of less than 10,000 individuals [24]. For example, the portion of the Navajo Nation located in Northern Arizona has a population density of approximately six people per square mile compared to the average U.S. population density of 345 people per square mile [22]. Many Diné families struggle economically (e.g., the Navajo Nation median household income = 25,963 USD, and 41 percent of households are below the federal poverty level) [25]. Diné children living in these communities may also be at an increased risk for developmental delays given the elevated environmental exposures to heavy metals including arsenic, uranium, and lead [26].

Though underdiagnosis and poor access to services have been broadly established for Indigenous children with autism [12,13], scant research has examined the reasons for this phenomenon—in particular, Diné parent/guardian experiences raising a child with autism including specific factors affecting their access to autism services [27,28,29]. Three prior qualitative studies (two unpublished) endeavored to better understand parent or service provider perspectives regarding Diné children with autism; however, these studies primarily relied on researcher observations to gather information with very small samples of less than 10 individuals. It is, therefore, critical to generate new knowledge on Diné parent/guardian experiences raising children with autism including factors affecting their access to autism services. This information will be valuable to the development of policy and programming intended to reduce health inequities for Indigenous communities, as autism is one of many chronic conditions affecting the health of individuals from childhood into adulthood. Because so little is known about this topic, this qualitative study sought to describe the lived experiences of Diné parents who have a child with autism. Specifically, from the perspectives of Diné parents/guardians of a child with autism, this study was intended to identify (1) the factors affecting access to autism diagnostic services, (2) factors influencing access to autism treatment services such as ABA therapy, and (3) areas for improvement in the system of care for Diné children with autism and their families across and around the Navajo Nation.

## 2. Materials and Methods

### 2.1. Study Design

This qualitative study was part of a larger study to adapt and pilot Parents Taking Action [30,31], which is an evidence-based, parent education and training program delivered by community health workers for Diné parents of children with autism. Approval to conduct this study was granted by the Northern Arizona University Institutional Review Board and the Navajo Nation Human Research Review Board. The study additionally adhered to any local or organizational approval processes required to recruit individuals in various communities in or around the Navajo Nation. A community advisory board (CAB) for the larger study that this qualitative study was part of, aided in participant recruitment and provided guidance on the interview protocol. The CAB included 13 individuals who had delivered autism services on the Navajo Nation or who themselves identified as a Diné parent of a child with autism.

### 2.2. Participant Characteristics

Individuals needed to meet the following eligibility criteria to participate in this study: be 18 years or older, identify as Diné, and be the parent or guardian of one or more children aged 2–12 years with autism. In total, 15 individuals participated in this study. One father was interviewed, and the rest were mothers. All identified as the child’s biological parent (for this reason we refer to participants as parents hereafter). The average age of participants was approximately 33 years (SD: 6.1; range: 24–46 years). Two parents (13.3 percent) were proficient (good or excellent) in speaking the Diné language, while the other 13 participants (86.7 percent) rated their Diné speaking skills as poor or fair. All participants reported not being proficient in writing the Diné language. Ten participants (66.7 percent) were married, living with someone, or engaged. Three participants (20 percent) were single or never married, and two participants (13.3 percent) were separated or divorced. High school was the highest level of education for nine participants (60 percent), four participants (26.7 percent) had some college education, one participant (6.7 percent) had a bachelor’s degree, and one participant (6.7 percent) had a master’s degree. Six participants (40 percent) were employed full-time, and nine participants (60 percent) were not employed. Nearly half (*n* = 7; 46.7 percent) of the participants had an annual household income of <25,000 USD, 26.7 percent (*n* = 4) of the participants had an annual income between 25,000 USD and 39,999 USD, and 13.3 percent (*n* = 2) of participants had 40,000–69,999 USD or 70,000 USD or more in annual household income. Only two (13.3 percent) of the participants rated their overall health as excellent. Most parents rated their health as good (*n* = 9; 60 percent), while three participants (20 percent) rated their health as fair and one (6.7 percent) as poor. Twelve participants (80 percent) had one child with autism, and the rest of participants (*n* = 3; 20 percent) had two children with autism. The average age of children with autism in the sample was 6.9 years (*SD* = 3.3; range 2–12 years). Most children with autism (*n* = 12; 80 percent) were male. All the children with autism were identified as Diné; two children were also identified as a member of another tribe; and one child was also identified as Latino, Hispanic, or Spanish. A majority of participants rated their child’s autism severity as mild (*n* = 9; 60 percent) and six parents rated their child’s autism severity as moderate or severe (40 percent). The average age of autism diagnosis for children was 3.2 years (*SD* = 1.6; range 1–8 years). All participant characteristics are displayed in Table 1.

### 2.3. Sampling Procedures

Participants were initially recruited from community-based organizations in and around the Flagstaff area (e.g., the Flagstaff Unified School District, local federally qualified health centers) and the Navajo Nation Western Agency Council area. Despite reaching out to each chapter and school in the Western Navajo Nation Agency area, recruitment numbers were low. Due to this, the recruitment area was expanded to the other Navajo Nation agencies (i.e., Central/Chinle, Northern/Shiprock, Fort Defiance, Eastern/Crownpoint; Figure 2). The study’s recruitment flyer was sent to all 110 chapters and 31 districts/schools on the Navajo Nation. This study was additionally given approval to recruit through the Navajo Nation Head Start and the Department of Diné Education. Recruitment efforts were most successful when a referral was directly received from this project’s CAB members. The referring entity first sought the parent’s approval to provide us with their contact information, and then a member of the study team contacted the parent by telephone and/or e-mail. Participant recruitment also occurred through a newspaper advertisement and through social media (i.e., Facebook). Of the 15 participants, 12 were referred by the project’s CAB members, three contacted the study team based on social media posts about the project, two were referred to the study team through the Navajo Nation Head Start, and one was referred from a Navajo Nation school district.

### 2.4. Measures

A semi-structured interview guide was developed with open-ended questions and directed probes guided by the study’s aims and based on past research related to ethnicity-based inequities in autism services access [6,7,8,9,10,11,12,13,14,15,16], as well as the constructs from the Modified Sociocultural Framework for Autism Health Services Disparities (Figure 1). Questions and probes were primarily about parents’ experiences accessing autism diagnostic and treatment services, as well as ways in which they thought that access to autism services could be improved for Diné children and their families on the Navajo Nation. The project’s CAB provided feedback on a draft of the interview guide before participant recruitment began. Their feedback was incorporated into the interview guide. Sample interview questions and directed probes are shown in Table 2.

### 2.5. Data Collection

Participant recruitment and data collection lasted approximately 13 months, from June 2021 to June 2022. After individuals expressed an interest in participating, a brief eligibility screener was completed over the telephone or by videoconference (i.e., Zoom). If determined eligible to participate, individuals were asked to provide verbal informed consent. Participants were then asked to complete a brief survey about their demographic characteristics and those of their child with autism. The survey was verbally administered by a Diné study team member by telephone or videoconference and took 15 to 30 min to complete. All survey data were entered into REDCap (Research Computerized Data Capture) tools hosted at Northern Arizona University. Participants were then given the option to continue with the semi-structured interview or to schedule another time for the interview. The interviews were also conducted by a Diné study team member and took most participants an hour to complete. All interviews were audio recorded and transcribed verbatim. Participants were mailed a 30 USD gift card following their interview.

### 2.6. Analysis

A directed content analysis approach [32] was used to analyze the interview data whereby the intention was to help extend and validate the Modified Sociocultural Framework for Autism Health Services Disparities Framework (Figure 2). In applying this approach, elemental coding methods were primarily used including descriptive, in vivo, and structural coding [33]. For example, under the structural code “Barriers to Accessing Autism Diagnosis Services” the code “Too Young” was used to capture information provided by participants regarding their experiences of being told by health professionals that their child was “too young” for them to diagnose autism. Two research team members independently coded each transcript. Coding discrepancies were resolved by consensus, and themes were distilled according to each study aim. All qualitative data analysis was performed in QSR NVivo [34].

## 3. Results

Twelve overarching themes were identified. These included four themes about factors affecting access to diagnostic services, five themes about access to treatment services, and three themes about ways to improve service access for Diné children with autism and their families. The main themes, including any subthemes and illustrative quotations, are presented according to each of the study’s three aims. Themes and subthemes are additionally described in relationship to the relevant social ecological levels (e.g., individual or family, interpersonal, community, health care system, physical or built environment, broader society) reflected in the Sociocultural Framework for Autism Health Services Disparities (Figure 1).

### 3.1. Factors Affecting Access to Autism Diagnostic Services for Diné Parents

Diné parents generally experienced difficulty accessing diagnostic services for children with autism. Factors affecting their access to diagnostic services were related to the sociocultural environment, the health care system, and the built or physical environment. In some cases, factors affecting access to autism diagnostic services cut across multiple social ecological levels (e.g., a parent seeking traditional healing for their child due to sociocultural beliefs about autism being caused by a taboo in the community and reinforced in their interactions with family and friends).

#### 3.1.1. Theme 1a: Despite Their Resilience and Resourcefulness, the Diagnostic Process Was Often Stressful and Emotionally Fraught for Diné Parents

Within this overarching theme, the following three subthemes were further identified regarding factors influencing parent experiences accessing diagnostic services for their child with autism: (1a1) parent awareness and understanding of autism affected access to diagnostic services, (1a2) parent denial about their child’s autism contributed to diagnostic delays, and (1a3) the beliefs of family and friends about autism influenced parents’ access to diagnostic services. The first two subthemes were most relevant to the individual and interpersonal levels, while the third subtheme spanned the individual, interpersonal, and community levels. Regarding the first subtheme, some parents discussed how they needed to seek out their own information about autism online and through other sources, to better understand autism as part of the diagnostic process (subtheme 1a1). In some cases, parents’ research on autism helped to expedite the diagnostic process. According to one parent:


*“I had already [done] some research on [autism], and she did show traits that I did see on YouTube, how autism kids were, and I did see it in her. And then that’s when I took her to IHS (Indian Health Service), and they referred us out to Phoenix, and that’s when they gave her, that she was diagnosed with autism.”*


Some parents experienced denial about their child’s autism, even if they already had some awareness and understanding of autism (subtheme 1a2). Furthermore, the beliefs and practices surrounding autism on the part of parents’ family members and friends affected their experiences accessing diagnostic services (subtheme 1a3). The cultural practice of prayer by parents’ family and friends was often viewed as supportive. One parent stated, “As far as cultural wise, that involvement, my family just prayed. They didn’t pray for a misdiagnosis or anything, they were just, they were involved prayer-wise”. Certain parents perceived the expression of these beliefs by their family and/or friends as an attempt to assuage their concerns about the child’s autism. For parents who were confident that their child had developmental issues, this could be frustrating and unhelpful. Parents additionally expressed how cultural beliefs about the causes of autism could make them feel responsible for their child’s condition and how related practices involving ceremonies and prayer were contrary to mainstream or Western medicine. One parent described this as follows:


*“In the Native culture, I was told that I had done something taboo-wise that’s affecting my child from speaking and the doctors were looking at it through science eyes. Whereas my family and medicine man was looking at it through different eyes, through spiritual beliefs. […] My Native side was telling me I did something wrong, or I went somewhere I shouldn’t have gone. That I just need to have prayers done and ceremonies done so my child will be okay. In science eyes, my child’s going to be like this his entire life. He may or may not speak. He’s going to have delays in his speech and in learning. Just listening to both sides was really hard.”*


#### 3.1.2. Theme 1b: Once Families Understood That Their Child Needed Autism Diagnostic Services, They Commonly Experienced Wait Times on the Magnitude of Months or Even Years

Five subthemes emerged regarding factors frequently contributing to the wait times experienced by Diné parents when they were trying to access diagnostic services for their children. These factors were related to the health care system such as difficulty obtaining needed referrals for diagnostic services (subtheme 1b1), heightened interaction with the health care system prompting access to diagnostic services (subtheme 1b2), and challenges scheduling appointments for diagnostic services (subtheme 1b3). Other factors were linked to the physical environment, particularly long distances to access autism evaluation services given the ruralness of where families lived in or around the Navajo Nation (subtheme 1b4). Last, broader societal-level factors including widespread COVID-19 pandemic-related service disruptions affected wait times for autism diagnostic services (subtheme 1b5). Regarding health care system barriers, some parents attributed their difficulty obtaining referrals for autism diagnostic services (subtheme 1b1) to limited clinician training and care coordination available through the Indian Health Service (IHS) and large academic medical centers with autism services relatively close to the Navajo Nation (e.g., the University of New Mexico, UNM). Certain parents had heightened interaction with the health care system due to their child’s concurrent conditions (e.g., epilepsy) and/or their child had used early intervention (EI) services, such as the Growing in Beauty EI program on the Navajo Nation, prior to the child receiving an autism diagnosis (subtheme 1b2). In these cases, more frequent exchanges with health professionals about their child’s development and functioning helped to prompt their access to diagnostic services. Several parents described how they encountered challenges scheduling their child’s diagnostic services due to their professional commitments and appointment availability (subtheme 1b3). For some parents, support from their family members helped them to mitigate this issue. Given the geographic isolation and population dispersion on and around the Navajo Nation, some parents experienced difficulty traveling to diagnostic services including the costs associated with traveling long distances (e.g., costs for gas, hotel lodging; subtheme 1b4). Several parents also described how it was challenging for service providers to visit them in a timely manner because they lived in a remote location. Conversely, closer geographic proximity to services (e.g., living in a border town such as Page or a large metropolitan area like Phoenix, often referred to as “the Valley”) made access to diagnostic services easier for several Diné families. The following quotes from two different parents help to illustrate these varying experiences:


*“I think around the reservation, it’s harder for Native Americans to get services and to get diagnosis and stuff. It gets harder for the Navajo Nation to do it […]. If they just have more people willing to work, everyone, it would help with the Navajo tribe. But I think in the Navajo Nation, a lot of workers out there aren’t willing to help and they’re not willing to understand.”*



*“[W]e had transportation, so it wasn’t difficult going there. We live in the Valley, so I think it was just like a commute downtown or something.”*


Because parents were interviewed during the COVID-19 pandemic, some parents also noted how broader service disruptions affected their access to diagnostic services by way of canceled appointments and communication breakdowns (subtheme 1b5).

#### 3.1.3. Theme 1c: After Parents Recognized That Their Child Had Developmental Issues and Raised These Concerns to Health Professionals, Limited Clinician Training and Cultural Humility Impeded Parents’ Access to Autism Diagnostic Services

The health care system factors related to clinician training and cultural humility were characterized by the following three subthemes: (1c1) parent was told that their child was “too young” for an autism diagnosis, (1c2) the child had a diagnostic shift from one developmental disability such as developmental delay to autism, and (1c3) parents experienced a lack of cultural humility (i.e., “a lifelong commitment to self-evaluation and self-critique, to redressing the power imbalances in the patient-physician dynamic, and to developing mutually beneficial and non-paternalistic clinical and advocacy partnerships with communities on behalf of individuals and defined populations” [19]), and in some instances discrimination by health professionals. Advocacy and perseverance helped several parents to overcome these barriers to accessing diagnostic services. Although most parents described how they had raised concerns about their child’s development to health professionals at an early age, some parents were told by health professionals that their child was “too young” for an autism diagnosis (subtheme 1c1). One parent explained how they viewed this as a missed opportunity for their child:


*“I’m his mom, so I know what he’s doing, what milestones he wasn’t hitting. I know from his behaviors that it was just different from my other kids. It was just mostly mainly just getting him diagnosed and they were just, like I said, he showed behaviors when he was one and a half and they couldn’t officially diagnose him just because they said he was too young. I think that was probably my biggest thing. He’s four now and he just got diagnosed officially. It’s like the doors opened up more because he officially is diagnosed when, if somehow it was diagnosed when he was younger, maybe he could have had a lot more interventions.”*


Other parents discussed how it was difficult for health professionals to determine if their child had autism. In these cases, initially the child was diagnosed with another developmental disability (subtheme 1c2). Discrimination and limited cultural humility shown by health professionals further impacted parents’ access to diagnostic services and their experience with the diagnostic process more generally (subtheme 1c3). Some parents attributed discrimination to their race. One parent described this as follows:


*“And the thing is here Snowflake, I don’t know if it’s still happening, it’s kind of a racial thing. They treat people of Color different. I told my mom, “I think it’s because they don’t want to help us.” And I told her, “Just don’t worry about it, wait till I come back.” As soon as I came back, I called… I found that paper and I called the doctor and then that doctor helped me. […] from what I could tell [about the] public school when it comes to kids with autism, from my experience, they don’t really do anything they just push them along.”*


Parents also described how some health professionals did not try to understand their culture. This made it challenging for them to communicate with these health professionals and to feel understood while accessing diagnostic services for their child. For certain parents, this experience was a deterrent from accessing diagnostic and subsequent treatment services for their child’s autism. This was not, however, always the case, insofar as clinician understanding about the family’s culture was mentioned by several families as a support while they accessed diagnostic services. One parent explained, “[A]s far as the provider, they were very understanding when it came to understanding ceremonies or the culture behind it”.

#### 3.1.4. Theme 1d: Factors Helping Diné Parents to Access Autism Diagnostic Services included Adequate Health Insurance, IHS Referrals, Care Coordination, Financial Aid to Travel, and an Efficient Evaluation Process

Nearly all Diné parents reported that their child’s health insurance covered the costs of most diagnostic services. Some families had their child enrolled in the Arizona Health Care Cost Containment System (AHCCCS) program, which is the state’s Medicaid program. One parent commented on this by stating, “Yeah, they [AHCCCS], again, pretty much covered 100% of the costs. I think even up to now, we haven’t paid out of pocket for anything. AHCCCS pretty much covers his medical costs”. Many parents also described how once they were able to access diagnostic services, the autism evaluation process was efficient. In addition, some parents described how they received financial travel support (e.g., reimbursement for hotel lodging) or transport services to take their child to diagnostic services. The support of health professionals in providing referrals for autism diagnostic services and coordination to these services was further mentioned by parents as a facilitator to accessing diagnosis for their child.

### 3.2. Results on Factors Influencing Access to Autism Treatment Services for Diné Parents

Five themes about factors influencing access to autism treatment services were identified. Similar to the factors identified as influencing access to diagnostic services, many factors cut across several social ecological levels. For many Diné parents, factors influencing their access to treatment services were dynamic, given the chronicity of their child’s autism and changing service needs over time. Additionally, some parents started receiving autism-related treatment services, such as services through an EI program, prior to when their child received an autism diagnosis.

#### 3.2.1. Theme 2a: Parent Perceptions of the Extent to Which an Autism Service Helped Their Child Affected Access

Many parents described the helpfulness of autism treatment services for their children in terms of core autism symptoms (e.g., social communication, challenging behavior). In some instances, parents would compare the utility of similar types of commonly used autism treatment services (e.g., speech and language therapy, occupational therapy) between providers and across service settings (e.g., school, hospital or clinic). Ultimately, parents viewed autism treatment services more favorably if they were able to observe positive change in their child’s core autism symptoms.

#### 3.2.2. Theme 2b: Social Support from Family and Friends Helped Parents Access Autism Treatment for Their Child

Although some parents and their families were new to understanding autism treatment, many parents reported that their immediate family members and friends were generally supportive of them accessing treatment services for their children. For some parents, the material support afforded by their family members and friends, including childcare or other types of financial support, helped parents to use autism treatment services. In describing support from her family, one parent stated, “They were willing to help in any way that they could. They were willing to babysit my other boys. They were willing to provide with […]. If I needed anything, they were willing to help me, like with gas money, or with lunch, or something”.

#### 3.2.3. Theme 2c: Obtaining Referrals and Care Coordination Services Affected Access to Autism Treatment

Within this theme related primarily to the health care system, three subthemes emerged related to the importance of parent advocacy (subtheme 2c1), lacking transition from EI (subtheme 2c2), and limited cultural humility on the part of autism treatment providers (subtheme 2c3). For some parents, advocacy was critical to helping them overcome challenges such as obtaining needed referrals or ensuring their child’s diagnosis met the requisite qualifying criteria in accessing autism treatment for their child (subtheme 2b1). In one case, a parent described having to seek out legal services to help ensure their child could access autism treatment services through a state agency. Not all children with autism in this study received EI services. However, for some who did, limited transition support from EI once the child turned or was close to turning three years old hindered their continued access to autism treatment services (subtheme 2b2). For some parents, this meant that their child did not receive any kind of autism treatment until they entered primary school at five or six years of age. Lacking cultural humility on the part of autism service providers further made accessing treatment more challenging for some parents (subtheme 2b3). Certain parents felt discounted in terms of their culture and related understanding about what they should be doing to help address their child’s autism symptoms. According to one parent:


*“I think there’s that lack of knowledge about where we come from and also our culture in itself. I think just generally […] they might have thought all Natives are the same, you know? I still kind of get that feeling from them. Yeah, and I think in the very beginning too, they would ask what type of services or what are we doing to help our child with his developmental delays? And I would say, ‘I don’t know, family time. We all get together and…’ This was before services, so I would say, ‘I don’t know, we as a family, we’re all there and supportive of each other. We try to do with what is available to us.’ And I think they would just kind of say like, ‘Oh, but that’s not a service.’ So, I just kind of felt like being invalidated [about] getting support from family […].”*


#### 3.2.4. Theme 2d: Costs of Autism Treatment Affected Access to Services

For many parents, health insurance, including AHCCCS (Arizona’s Medicaid program) and Supplemental Security Income (SSI), helped to cover their child’s treatment expenses. One parent noted “[…] His support and insurance grew because he was just really on AHCCCS [AZ Medicaid], and now he’s on Arizona Long Term Care. He was approved for [it] because of his severe autism”. Nevertheless, some parents were unable to access certain types of autism treatment including parent-mediated intervention training due to the high out-of-pocket costs.

#### 3.2.5. Theme 2e: Availability of and Geographic Proximity to Services Impacted Treatment Access

Within this theme, the following three subthemes were identified: (2e1) distance to services was prohibitive for parents with limited resources, (2e2) availability of autism treatment at the child’s school and in their community facilitated use, and (2e3) the COVID-19 pandemic caused widespread disruptions in access to autism treatment. These factors were primarily rooted at the community and societal levels and pertain to the physical environment and the health care system. Many autism treatment services that parents learned of were based in cities requiring travel and additional costs, which was prohibitive for some Diné parents, especially those with limited resources (subtheme 2e1). One parent elaborated that in addition to the distance required to travel to the service, the frequency of treatment service was also a barrier to accessing it. Some parents were able to travel to treatment services; however, the costs associated with this were considerable for them. As one parent explained:


*“Sometimes, we had to reschedule because traveling there was on us, sometimes we didn’t have a reliable vehicle to make it out to Flagstaff and back. Then of course you travel that far, you got to […] get food. Then, sometimes, we had to do lodging because of the time. Maybe they wanted to see us at 8 a.m. Well, that would cause us to leave Window Rock at like 4:30 in the morning. We would probably have to go out there and get a hotel room and that was on us.”*


In a few cases, parents moved to a larger city so their child could access autism treatment. One parent described this as follows:


*“I’m very fortunate, and this is what I mean. I don’t understand how other parents are doing [it] because we were very fortunate to be educated, to have that ability to move him to the city, and get services, and get him diagnosed, and then to have a professional school who is willing to work with you and figure things out to support you.”*


Given these challenges with autism treatment availability and distance to services, many parents expressed how having autism treatment services available at the child’s school and/or in the family’s community facilitated access (subtheme 2e2). Some parents primarily relied on the child’s school for autism treatment services, while others relied on clinics or hospitals in their community. Again, similar to the challenges that parents experienced accessing diagnostic services, the COVID-19 pandemic caused considerable disruptions to access to treatment services (subtheme 2e3). These service disruptions were often compounded by limited social support and increased caregiving demands due to school closures.

### 3.3. Results on Ways to Improve Access to Services for Diné Children with Autism

Three themes were identified relating to ways that the system of care for children with autism and their families could be improved in and around the Navajo Nation in the future. Themes were related to autism awareness, autism-focused support groups, and increased availability and quality of autism services for Diné families. Themes pertained to the family and interpersonal levels, as well as the health care system and physical environment.

#### 3.3.1. Theme 3a: Greater Awareness and Understanding of Autism among Diné Families and Communities Could Improve Access to Services

Many parents expressed how other Diné families may not recognize developmental issues related to autism in children, delaying their access to services. Due to this, parents indicated how efforts to raise autism awareness within communities are needed. Along these same lines, several parents discussed how more tailored autism informational resources for Diné families are needed. Some suggested a centralized online repository of autism information or a helpline where families can seek assistance navigating the system of care for autism. One parent elaborated:


*“How could they improve it? I would say, if there was a whole website, or a whole pamphlet or brochure or so, of numbers and who to call or so, I think that would be awesome. That for sure would’ve helped me. And then, how to go about getting evaluated, and then getting the services. I think that would be very helpful.”*


#### 3.3.2. Theme 3b: Diné Families May Benefit from Autism-Focused Support Groups

Several parents suggested that groups for parents and other family members of children with autism would be helpful in building social connections and helping parents address questions. According to a parent, “[S]upport groups maybe with other parents because just bouncing ideas off each other would help us quite a bit”.

#### 3.3.3. Theme 3c: Increased Availability and Quality of Autism Services across and around the Navajo Nation Is Needed

Within this theme, the following four subthemes emerged: (3c1) more intensive and higher quality autism services are needed, (3c2) increased provider training on autism and care coordination is needed, (3c3) greater assistance with major service system transitions would be beneficial, and (3c4) flexibility in scheduling appointments for services would be helpful to some parents. One parent who had moved away from the Navajo Nation further described the root of many challenges in terms of the rural context.


*“I think just more availability because I can’t imagine being in a remote location on the reservation and having to take your child like an hour away for an hour service and then coming back and just coordinating all of that. So, I hope more locations do pop up. I don’t know what’s currently available for families on the reservation right now, but I can only imagine. I think my family kind of has it easy in terms of being here in the Valley where our nearest service is 10 miles away and there’s a lot of resources available and I just hope one day that’s the same for families back home. It’s like they could just go around the corner or something to get their service.”*


Some parents suggested that more intensive and/or higher quality services are needed for Diné children with autism in and around the Navajo Nation (subtheme 3c1). Parents described how there are often no autism treatment providers in towns on the Navajo Nation or how they must travel a significant distance to access services in a different town for their child’s autism. Parents also expressed that even when services become available, they are not always delivered by qualified professionals (e.g., students in training programs), which may perpetuate mistrust and skepticism about the potential benefits of the services. Relatedly, parents described the importance of increased provider training about autism and care coordination processes that support service access (subtheme 3c2). Greater assistance with major service transitions (e.g., the transition from EI) was further described as potentially beneficial for Diné families (subtheme 3c3). One parent expressed, “I wish they didn’t age out, [and] have the program where the program just stops, and you don’t have anywhere else to go or continue the services. Once [the child] hit four”. Lastly, flexible scheduling for autism services to better accommodate parents’ availability was mentioned as helping to potentially facilitate access (subtheme 3c4). One parent explained, “I think just working with the parent schedule. Like today, it wouldn’t have worked out if you said, “Can we meet at 9:00?” I think just working with the parent schedule would help. Some moms are so busy, they’re free […] when their child’s napping or something”.

### 3.4. Connections between Themes on Access to Autism Services for Diné Parents

The 12 themes identified were connected. The four themes about diagnostic services access were related to the five themes about treatment services access, and these themes on access to diagnostic and treatment services were together related to the three themes on ways to improve access to autism services collectively (i.e., diagnosis and treatment services) across and around the Navajo Nation. These thematic connections were mapped to show how they may relate in terms of the potential pathways and experiences of accessing autism services, as well as how they may be examined in future work to improve access to autism services for Diné families (Figure 3). In summary, difficult experiences accessing diagnostic services often contributed to subsequent or concurrent challenges accessing treatment services such as behavioral therapy for children with autism. For some children, access to autism-related services began through an EI or Head Start program prior to their autism diagnosis. Depending on how the transition from these programs went, diagnosis might or might not have occurred, affecting the continuation of access to autism treatment services once the child aged out of EI or Head Start. Due to this, the relationship between access to diagnostic and treatment services is depicted as bidirectional versus only moving linearly from diagnosis to treatment. Similarly, improvement areas for access to autism services stem from and would be likely to impact access to both diagnosis and treatment services for Diné families. The improvement areas highlighted by the three themes identified in this study underscore the multilevel nature of factors affecting access to autism services and the solutions that will be needed to address the intersections of culture, poverty, and ruralness that many Diné families are affected by while raising a child with autism.

## 4. Discussion

This study is one of the first to describe the lived experiences of Diné parents who have a child with autism and to identify specific determinants of families’ access to diagnostic and treatment services, as well as ways that the system of care for children with autism and their families can be improved across and around the Navajo Nation. The results revealed that barriers restricting access to autism diagnostic and treatment services were present across social ecological levels (i.e., parent or family, interpersonal, community, health care system, larger physical or built environment, societal) [35]. The barriers that most commonly impeded families’ access to diagnosis and treatment services included long wait times to receive services, inadequate training on autism among health professionals, clinician bias and lack of cultural humility, trouble scheduling appointments, difficulties traveling long distances to services, and the widespread disruption in access to services caused by the COVID-19 pandemic. Barriers related to the physical and built environment (e.g., long distances to autism services) and the health care system (e.g., clinician bias and lack of cultural humility) were common for both access to treatment and diagnosis. Sociocultural factors (e.g., cultural beliefs about the child’s autism being caused by a taboo), however, were mostly discussed as barriers inhibiting access to diagnostic but not treatment services. For many Diné parents, support from family and spiritual or community groups (e.g., family prayer, family and friends helping with childcare while the parent took their child to autism service appointments) were facilitators to accessing autism diagnostic and treatment services.

Barriers in the physical and built environment that frequently affected access to autism services for Diné parents (e.g., difficulties traveling long distances to services) are common for families living in rural communities due to geographical isolation. These challenges are often exacerbated by poverty and related issues such as not having personal transportation or the financial resources to afford hotel lodging in a city. Other structural issues in the physical and built environment for families living in or around the Navajo Nation and other rural areas, such as the digital divide, poor public roads, and other transportation infrastructure, further contribute to inequities in autism services access. Our findings also highlight the limited availability of specialized autism services (e.g., ABA therapy) on the reservation. Specialized therapy services for autism were often reported to be hours away, which again made it difficult for families to access these services for their child. Strikingly, some participants reported having made the decision to move away from the Navajo Nation to a larger metropolitan area, such as Phoenix, Arizona, so they could more easily access evidence-based treatment services, such as ABA therapy, for their child with autism.

Barriers at the health care system level were also found to add significant stress for Diné parents raising a child with autism. Parents in this study attributed late diagnosis to (1) providers not being concerned about their child being at risk for autism and not being certain of an autism diagnosis when children are young, (2) providers not making referrals for autism evaluation services, and (3) long wait times to receive autism diagnostic services. Some parents reported that they knew something was different about their child’s development but clinicians were dismissive or did not recognize the signs of autism when the parents had voiced concerns about their child’s development. These issues may be related to a lack of training on autism symptoms or not having sufficient experience to be comfortable screening, referring, and/or diagnosing autism. Parents also reported difficulties with uncoordinated and unsupportive systems, as well as what they perceived as discriminatory practices of providers. In one instance, for example, a parent reported resorting to legal assistance to obtain autism treatment services for their child. The disparities in diagnosis of children with autism are consistent with what has been reported in the literature [8,36,37,38,39,40]. Aylward et al. reported that culturally and linguistically diverse and low-income children often do not receive autism-specific services during important developmental windows (e.g., ages 0–3 years) leading to racial, ethnic, and socioeconomic disparities in access to diagnosis, intervention, and other services for children with autism [38]. Although the out-of-pocket costs of some autism treatment services, such as parent training programs, were reported to be prohibitive by parents, for most parents in this study adequate insurance coverage through the Indian Health Service and AHCCCS (Arizona’s Medicaid program) helped support them in receiving an autism diagnosis for their children. In addition, the availability of certain treatment services, such as speech and language therapy and occupational therapy through EI and schools, helped ensure that children were able to access some autism treatment services.

Sociocultural factors reported in prior research on autism in other culturally and linguistically diverse populations [16,36,37,38,39,40] were reflected in the parents’ narratives from this study. These included parents’ concerns about possible stigmatization by their family members and the broader community related to the child’s autism, lack of awareness about developmental disabilities, and limited understanding of autism symptoms and treatment services [38]. For example, several parents described feeling alone in trying to understand autism and a feeling of frustration in not receiving support for timely diagnostic and treatment services. Parents also reported the emotional impact of their child’s autism diagnosis—and in some cases denial about their child’s autism—because of the great magnitude of change that would be needed to best support their child. While the lack of awareness about disability and autism was discussed as a barrier in receiving diagnosis, many parents highlighted the value of support provided by their family and community members after the child received an autism diagnosis. These findings reflect the strengths of Diné parents and the importance of their community support. Some parents expressed taking on the role of advocacy for services, particularly when they experienced difficulty accessing services prior to receiving a diagnosis. Parents also reflected on the impact of their Native spirituality and culture in coping with the demands of having a child with autism, and on their children to incorporate Diné teachings to help shape their identity (e.g., Know your children, who they are, their weakness, and their strongest point. Focus on the child’s weakness to support them.). These principles of advocacy, self-determination, and empowerment reflect concepts of K’e (kinship), Hózhó (harmony and balance), and the Diné philosophy “taa whi aji at’eego deeya nizhonigo hiji naado” (self-empowerment). These are part of the Hózhó Resilience Model that has been developed to guide health-related research with Diné communities (Figure 4; [41]).

### 4.1. Recommendations

This study’s results highlight the need for raising awareness at the community level about developmental milestones, early signs and symptoms of autism, and culturally responsive early intervention. Additionally, the study’s findings support the need for providing greater training to clinicians about autism symptoms and early identification, as well as cultural responsiveness in their practice serving children and families across and around the Navajo Nation. More frequent developmental screening at well child visits and/or in childcare settings via commonly used screeners, such as the Ages and Stages Questionnaire, and repeated autism screening may also be beneficial to improving early autism diagnosis among Diné children. The Project Extension for Community Healthcare Outcomes (ECHO), which has improved healthcare delivery in other rural areas of the United States and has been applied to training pediatric clinicians about autism, may also hold promise in terms of increasing early autism identification in primary care settings. Some efforts have already been made to train pediatricians in IHS clinics to identify autism at earlier ages in children through the routine use of first level (M-CHAT) and second level (Screening Tool for Autism in Toddlers and Young Children) autism screening tools. Similarly, some medical centers in the Navajo Nation have identified local diagnosticians who they can easily refer children to for an autism evaluation following a positive autism screen. Further, continued efforts are needed to improve service delivery by using technological innovations such as telehealth services. These types of initiatives may dovetail with larger efforts to improve rural infrastructure related to broadband and roads across the Navajo Nation. This study’s results also point to the need for creation of a centralized and easily accessible resource center for Diné families who have children with autism or other developmental disabilities to help families find all information and resources available to them on and off the reservation. Moreover, the creation of and maintenance to sustain partnerships between various programs that serve children on the Navajo Nation may help to reach and more comprehensively serve more Diné families who have children with autism or other developmental disabilities. For example, the Navajo Nation Early Childhood Collaborative (Project I-LAUNCH) has in recent years brought together various entities and other stakeholders to collaboratively plan and discuss initiatives and advance community education focused on young children across and around the Navajo Nation. Last, the development and/or adaptation of evidence-based parent education and training programs, such as Parents Taking Action [30,31], to build family and community capacity by way of community health workers (other parents or family members who have a child with autism) may aid in increasing autism awareness and access to services. For instance, it may be possible to embed parent education and training programs into existing programs, such as EI and Head Start, which already deliver services to young children and their families in the Navajo Nation. There may also be potential for community health representatives who work in clinics or other community health worker-driven programs serving families with children to deliver parent education and training programs to those families who have been identified as having a child at risk for autism or other developmental issues. These and other efforts may better support Diné families, particularly during periods of service transition such as the transition out of EI when a child turns three years old.

### 4.2. Limitations

This qualitative study’s findings contribute to the dearth of research on Indigenous parents’ experiences raising children with autism, in particular factors affecting access to services. Several key limitations should be noted. As this was a qualitative study, the sample size was small, even though it was larger than samples in previous studies on related topics. For this reason, and similar to many other qualitative research studies, this study’s findings are not intended to be generalizable. Rather, this study’s findings provide new information on a phenomenon for which little is known and may serve as a starting point for future research involving larger samples and quantitative methods. In addition, some research suggests that thematic saturation (i.e., the inability to identify new themes) can occur in 6 to 12 twelve interviews [42]. Due to the various recruitment challenges brought about by the COVID-19 pandemic, this study’s sample likely included better resourced Diné parents who were able to participate despite the immense hardship the pandemic incurred, especially initially for the Navajo Nation. Indeed, this study’s sample had higher socioeconomic status, including education attainment, compared to that reported for families living across Navajo Nation: approximately 40 percent of families in the current study had some college education compared to only 7 percent reported for families across the Navajo Nation [24]. Thus, parents in this study may have had more awareness about autism compared to parents living in more rural areas of the Navajo Nation. Future research may therefore consider a larger and more representative sample of Diné parents of children with autism that includes those living in rural areas of the Navajo Nation. Along these same lines, it may be helpful to capture the experiences of Diné parents who have autistic youth and/or young adults to better understand similar and unique factors affecting their access to services. Last, because our study occurred during the COVID-19 pandemic it is impossible to disentangle how much the pandemic in and of itself versus other factors impacted parents’ experiences accessing autism services.

## 5. Conclusions

This qualitative study provides new knowledge on Diné parents’ lived experiences raising children with autism. Critical factors impacting access to autism diagnosis and treatment services along with ways to improve the system of care for Diné parents are reflected by the 12 themes that were identified. This study’s findings collectively reinforce the resilience and resourcefulness of Diné parents that have a child with autism, while highlighting several barriers that may be addressed through programmatic and policy changes. These changes may include initiatives to bolster rural infrastructure (e.g., increased broadband and free hotspots, improved highway systems), thereby facilitating easier access to telehealth and physical autism services across and around the Navajo Nation. In addition, the use and adaptation of existing clinician training programs on autism (e.g., how to identify autism in primary care) to better understand central aspects of Diné culture (e.g., traditional healing, Hózhó) more specifically may be beneficial to reducing some barriers that parents commonly encounter in accessing diagnostic and treatment services through the health care system. Autism treatment programs that employ delivery models consistent with or are embedded within existing health education and promotion programs, such as the Community Health Representative program, using community health workers, may also hold promise for advancing autism treatment accessibility for Diné families. Greater coordination and collaboration between existing programs (e.g., EI, Head Start, schools, IHS) serving children with autism and their families across the Navajo Nation through initiatives, such as the Navajo Nation Early Childhood Collaborative (Project I-LAUNCH) and the Navajo Nation Autism Community Referral Group, may further create a more robust and effective system of care. By addressing these determinants, advances in health equity are possible for Diné families and may be possible to apply and adapt with other Indigenous communities.

## Figures and Tables

**Figure 1 ijerph-20-05523-f001:**
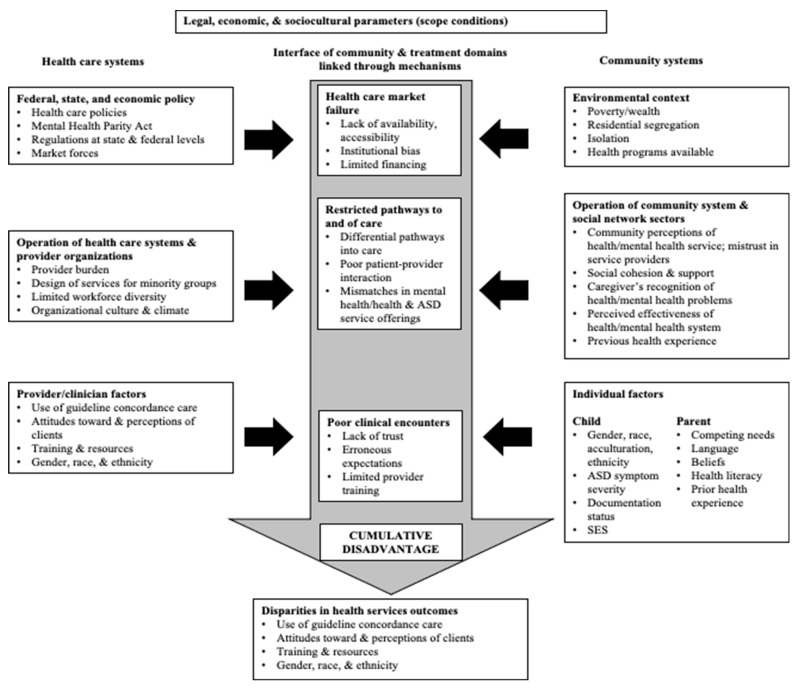
Modified Sociocultural Framework for ASD Health Services Disparities [15].

**Figure 2 ijerph-20-05523-f002:**
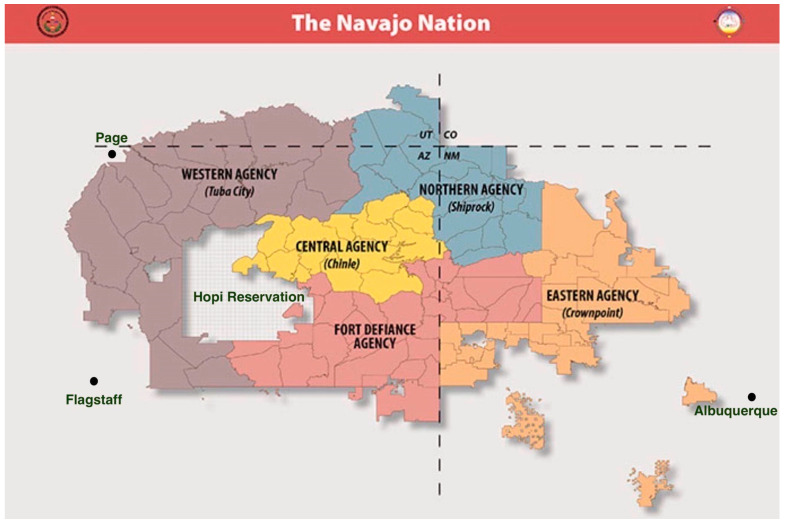
Map of the Navajo Nation with the Agency Councils [23].

**Figure 3 ijerph-20-05523-f003:**
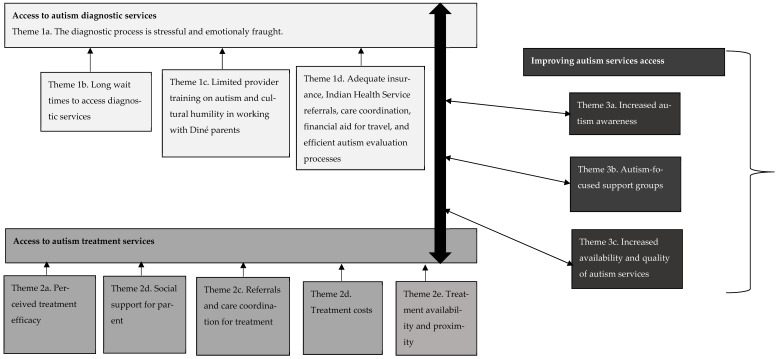
Connections between Themes on Access to Services for Diné Parents of Children with Autism.

**Figure 4 ijerph-20-05523-f004:**
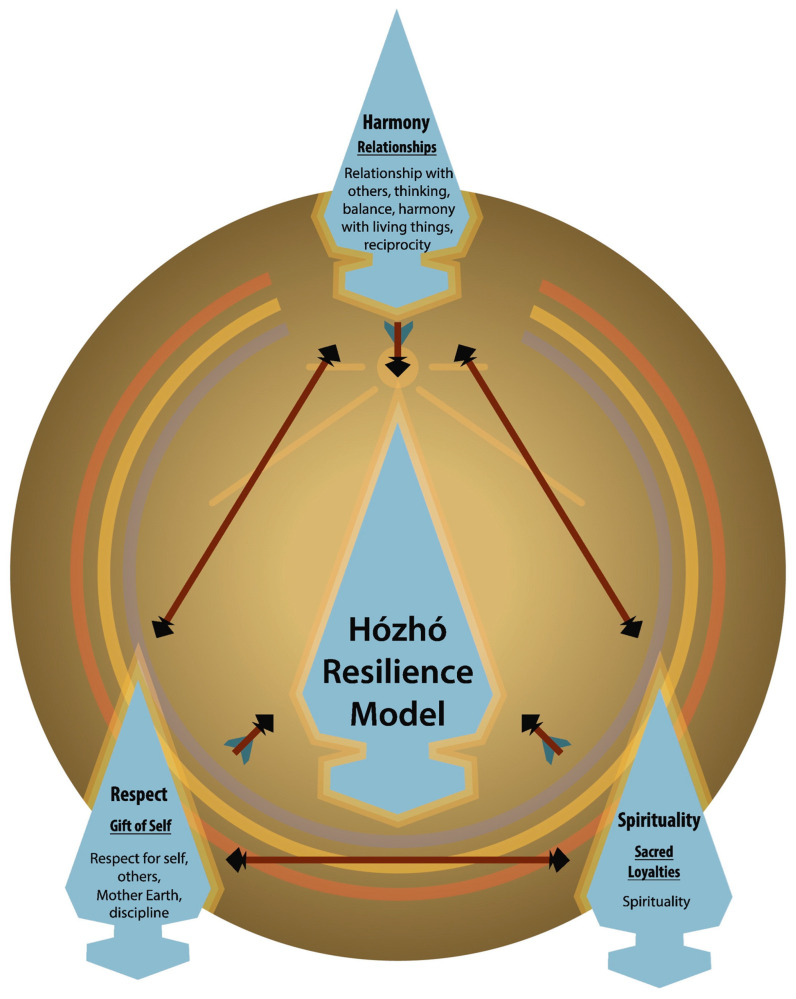
The Hózhó Resilience Model [41].

**Table 1 ijerph-20-05523-t001:** Diné Interview Participant Characteristics.

	*n*	% or *M (SD)*
Parent Characteristics		
Relationship to the child with autism		
Mother	14	93.3
Father	1	6.7
Gender		
Female	14	93.3
Male	1	6.7
Age, years	15	33.3 (6.1)
Identifies as Diné	15	100
Identifies as member of another tribe	0	0
Identifies as Latino	0	0
Verbal proficiency in the Diné language		
Poor or Fair	13	86.7
Good or Excellent	2	13.3
Written proficiency in the Diné language		
Poor or Fair	15	100
Good or Excellent	0	0
Marital status		
Married, living with someone, or engaged	10	66.7
Single or never married	3	20
Separated or divorced	2	13.3
Highest level of education		
High school	9	60
Some college	4	26.7
Bachelor’s degree	1	6.7
Master’s degree or doctorate	1	6.7
Employment status		
Employed full-time	6	40
Employed part-time	0	0
Not employed	9	60
Household income level annually		
<USD25,000	7	46.7
USD25,000–39,999	4	26.7
USD40,000–69,999	2	13.3
USD70,000 or more	2	13.3
Overall health status		
Poor	1	6.7
Fair	3	20
Good	9	60
Excellent	2	13.3
Number of children with autism		
One child	12	80
Two children	3	20
Child Characteristics		
Age, years	15	6.9 (3.3)
Gender		
Male	12	80
Female	3	20
Identifies as Diné	15	100
Identifies as member of another tribe *	2	13.3
Identifies as Latino	1	6.7
Autism severity		
Mild	9	60
Moderate or Severe	6	40
Age of autism diagnosis, years	15	3.2 (1.6)

Note: Abbreviations: *M*, mean; *SD*, standard deviation. * Other tribes were Hopi or Sioux.

**Table 2 ijerph-20-05523-t002:** Example Interview Questions and Directed Probes about Diné Parents’ Lived Experiences Raising a Child with Autism, by Study Aim.

Study Aim	Example Interview Questions	Possible Directed Probes
1. Factors affecting access to autism diagnostic services	Thinking back to the time that your child was diagnosed with autism, what were the biggest obstacles you experienced getting your child diagnosed?	Was this experience stressful? Why or why not? How did your understanding of autism influence your child’s diagnosis? Were costs or insurance coverage an issue for you in getting the diagnosis?
2. Factors influencing access to autism treatment services	Thinking about the services that you have used for your child’s autism, what has helped you get these services?	Did someone give you the referrals that you needed to get your child’s autism services? Did someone help you to coordinate your child’s autism services? Were services offered at your home or somewhere nearby?
3. Areas for improvement in the system of care for Diné children with autism and their families on the Navajo Nation	How do you think that access to autism services could be improved for Navajo families?	For example, more autism public awareness or education campaigns? Or more availability of autism services on the Navajo Nation?

## Data Availability

Qualitative and survey data for this study belong to the Navajo Nation Human Research Review Board.

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
