# Peer review of "“Know Your Children, Who They Are, Their Weakness, and Their Strongest Point”: A Qualitative Study on Diné Parent Experiences Accessing Autism Services for Their Children"

_ijerph, 2023, doi:10.3390/ijerph20085523_

Round 1
Reviewer 1 Report (Previous Reviewer 2)
Excellent paper. This will be a great resource to those who work closely with Diné families. I think it is also a starting point for how we can improve resource access for Diné individuals with ASD.
Well done.
This manuscript is a resubmission of an earlier submission. The following is a list of the peer review reports and author responses from that submission.
Round 1
Reviewer 1 Report
This manuscript deals with an important topic. But, there are some concerns exist as follows:
1. The presentation of qualitative data and the small sample size weaken the study's outcome to a great extent.
2. The presentation of the manuscript seems like a report rather than a scientific paper, especially with many quotations from the parent's answers.
3. The abstract is very long.
4. The writing style should be formal from the third-person perspective. Do not use we (E.g., line 10 in the abstract )
5. Table 1: M (SD) refers to what? The full term should be presented in the footnote.
6. Line numbering is highly recommended to be added to facilitate the reviewing process.
Reviewer 2 Report
This is a well-written paper with excellent details on design and analysis. I think there is more potential for the conclusion section. You present the barriers to ASD diagnosis and potential solutions. What do you think could work regarding improved rural infrastructure? Are you referring to more telehealth services? What could the collaboration with IHS or CHR look like? The way it reads to me seems like you are proposing the training of CHRs to help facilitate access to services? Are there any existing models that work on the Navajo Nation? For example, training of pediatricians to conduct ASD screenings for preliminary diagnosis, such as what takes places at Gallup IHS. Or how pediatricians work with PHNs at Northern Navajo Medical Center to refer families to a local private child psychologist for quicker diagnosis. These models still will benefit from improvement, but some of these innovative practices may hold promise.
More details on how effective collaboration amongst programs on the Navajo Nation would be helpful. I think many reading this paper are going to be looking for potential solutions. I know I would like to see these.
